# Comparative Transcriptome and sRNAome Analysis Suggest Coordinated Citrus Immune Responses against Huanglongbing Disease

**DOI:** 10.3390/plants13111496

**Published:** 2024-05-29

**Authors:** Muhammad Saqib Bilal, Gomes Paul, Ze Yu, Le Xu, Tang Cheng, Baoping Cheng, Muhammad Naveed Aslam, Ayesha Baig, Hongwei Zhao

**Affiliations:** 1Key Laboratory of Plant Immunity, College of Plant Protection, Nanjing Agricultural University, Nanjing 210095, China; muhammadsaqib145@njau.edu.cn (M.S.B.); gomes.paul.46@gmail.com (G.P.); 2019202024@njau.edu.cn (Z.Y.); 2020202033@stu.njau.edu.cn (L.X.); 2021202038@stu.njau.edu.cn (T.C.); 2Institute of Plant Protection, Guangdong Academy of Agricultural Sciences/Key Laboratory of Green Prevention and Control on Fruits and Vegetables in South China, Ministry of Agriculture and Rural Affairs/Guangdong Provincial Key Laboratory of High Technology for Plant Protection, Guangzhou 510642, China; 3Department of Plant Pathology, Faculty of Agriculture and Environment, The Islamia University of Bahawalpur, Bahawalpur 63100, Pakistan; naveed.aslam@iub.edu.pk; 4Department of Biotechnology, COMSATS University Islamabad Abbottabad Campus, Abbottabad 22010, Pakistan; ayeshabaig@cuiatd.edu.pk

**Keywords:** lemon, Shatangju, starch accumulation, antioxidant enzymes, DE-miRNA, csi-miR399, csi-miR166

## Abstract

Citrus Huanglongbing (HLB), caused by the phloem-inhibiting bacterium *Candidatus* Liberibacter asiaticus (*C*Las), is the most devastating citrus disease, intimidating citrus production worldwide. Although commercially cultivated citrus cultivars are vulnerable to *C*Las infection, HLB-tolerant attributes have, however, been observed in certain citrus varieties, suggesting a possible pathway for identifying innate defense regulators that mitigate HLB. By adopting transcriptome and small RNAome analysis, the current study compares the responses of HLB-tolerant lemon (*Citrus limon* L.) with HLB-susceptible Shatangju mandarin (*Citrus reticulata* Blanco cv. Shatangju) against *C*Las infection. Transcriptome analysis revealed significant differences in gene expression between lemon and Shatangju. A total of 1751 and 3076 significantly differentially expressed genes were identified in Shatangju and lemon, respectively. Specifically, *C*Las infected lemon tissues demonstrated higher expressions of genes involved in antioxidant enzyme activity, protein phosphorylation, carbohydrate, cell wall, and lipid metabolism than Shatangju. Wet-lab experiments further validated these findings, demonstrating increased antioxidant enzyme activity in lemon: APX (35%), SOD (30%), and CAT (64%) than Shatangju. Conversely, Shatangju plants exhibited higher levels of oxidative stress markers like H_2_O_2_ (44.5%) and MDA content (65.2%), alongside pronounced ion leakage (11.85%), than lemon. Moreover, microscopic investigations revealed that *C*Las infected Shatangju phloem exhibits significantly more starch and callose accumulation than lemon. Furthermore, comparative sRNA profiles revealed the potential defensive regulators for HLB tolerance. In Shatangju, increased expression of csi-miR166 suppresses the expression of disease-resistant proteins, leading to inadequate defense against *C*Las. Conversely, reduced expression of csi-miR166 in lemon plants enables them to combat HLB by activating disease-resistance proteins. The above findings indicate that when infected with *C*Las, lemon exhibits stronger antioxidative activity and higher expression of disease-resistant genes, contributing to its enhanced tolerance to HLB. In contrast, Shatangju shows lower antioxidative activity, reduced expression of disease-resistant genes, significant ion leakage, and extensive callose deposition, possibly related to damage to plant cell structure and blockage of phloem sieve tubes, thereby promoting the development of HLB symptoms.

## 1. Introduction

Citrus Huanglongbing (HLB), formerly known as ‘Citrus Greening’, caused by *Candidatus* Liberibacter asiaticus (*C*Las), a genus of phloem-inhibited bacterium, which is transmitted by the Asian citrus psyllid (ACP), is debatably the most damaging disease intimidating citrus production worldwide [1]. Citrus HLB is characterized by yellowing of early shoots, mottling of fully grown old leaves, or a zinc deficiency-like appearance in mature fresh leaves [2]. In 1919, the first incidence of HLB was reported in China’s Guangdong province [3]. Despite optimistic scientific progress, HLB posed an unprecedented threat to the Chinese citrus industry. The Ministry of Agriculture and Rural Affairs of China listed it as one of the top ten crop diseases in 2020 [4].

Currently, no commercially utilized citrus varieties are resistant to HLB. However, only a few HLB-tolerant citrus verities can remain economically viable for 5–10 years after *C*Las infection [5]. The most economical and environmentally friendly method for curbing the spread of HLB is selective breeding [6]. Accessibility to adequate resistance genes is crucial to the effectiveness of this technique. Unfortunately, in citrus and its close relatives, no HLB resistance gene has been found yet. Thus, there has not been much development in breeding for HLB resistance. However, certain citrus intergeneric hybrids have been approved as rootstocks and have demonstrated HLB tolerance [7].

Gene function can be disrupted in several ways, involving CRISPR/Cas-mediated gene editing, T-DNA insertion, and mRNA-targeting [8]. MicroRNAs (miRNAs) are groups of small, approximately 19~24-nucleotide-long noncoding RNAs [9]. miRNAs can adversely regulate gene expression via translational inhibition or cleavage, playing a critical role in different plant biological and metabolic processes [10]. In contrast to alternative gene expression regulators like transcription factors (TFs), genes targeted by miRNAs are amenable to bioinformatics-based prediction and experimental validation. Thus, concurrent comparison between clients and regulators is feasible.

To better comprehend the HLB-tolerance mechanisms in citrus, comparative transcriptome profiling of susceptible and tolerant citrus cultivars in response to *C*Las infection has been conducted using RNA-seq and other high-through sequencing technologies [11,12,13]. Transcriptomic research demonstrated a more robust immune response in tolerant rough lemon (*Citrus jambhiri*) compared with susceptible sweet orange (*Citrus sinensis*) upon *C*Las infection [14]. Genes associated with cell wall metabolism, lipid metabolism, and signaling exhibited an integral role in the resistance of Kaffir lime (*Citrus hystrix*) and Mexican lime (*Citrus aurantifolia*) to HLB [15]. An extensive transcriptomic study of *Citrus trifoliata*, *Citrus sunki*, *Citrus sinensis*, and three different hybrids revealed a distinct genetic pathway for HLB tolerance involving the suppression of gibberellin (GA) production and the promotion of cell wall reinforcement [16]. Recently, our research group discovered that citrus can detect *C*Las infection and induce HLB immune responses. This response can be boosted by natural and exogenous methyl salicylate (MeSA) application. We showed transgenic and multiyear trial evidence suggesting MeSA is a strong community immune signal [17]. Comprehensive multi-omic research on citrus varieties against *C*Las infection revealed that few defense and cellular responses were comparable across HLB-tolerant citrus cultivars, and a few were specific to certain cultivars or genotypes.

Breeding HLB-tolerant citrus varieties is a promising long-term strategy to control HLB. Further research is needed to comprehend the variations in host response across tolerant and susceptible citrus cultivars, which can offer valuable insights for developing HLB-resistant or tolerant varieties in the future [18]. In this study, by using transcriptome and sRNAome profiling, we analyzed the defensive mechanisms and genetic regulation of HLB-tolerant lemon and susceptible Shatangju mandarin. We selected Shatangju due to its prominence as a citrus cultivar in South China. China’s Agricultural yearbook editorial board estimated that 491,105 hectares were designated for cultivating Shatangju in 2021 [19]. In this study, we rigorously investigated the response of two distinct citrus genotypes (*C. limon* and *Citrus reticulata* Blanco cv. Shatangju) to HLB by comprehensive transcriptomic investigation. We also identify potential defensive regulators for HLB tolerance through integrative transcriptome and sRNAome analysis of both lemon and Shatangju. The primary aim was to evaluate and compare their respective tolerance levels to HLB. Our findings revealed that *C. limon* exhibited notable tolerance to HLB, whereas Shatangju displayed heightened sensitivity to the disease.

## 2. Results

### 2.1. Lemon Exhibits More Pronounced Resistance to CLas Infection Than Shatangju

Lemon leaf samples collected after 14 weeks of inoculation did not possess a typical blotchy mottle appearance. However, 24 weeks after inoculation, lemon plants showed minimal growth retardation and continued growth of fresh branches. Meanwhile, older leaves were more likely to have a regular blotchy mottled pattern (Figure 1A). On the contrary, 12 weeks after inoculation, most Shatangju plants began to show symptoms on their new leaves and their growth was considerably stunted. During the subsequent 6 months, these symptoms progressed to severe, characterized by blotchy mottling and yellowing of the midrib (Figure 1B). To compare the difference in *C*Las growth, the presence of *C*Las was quantified by using qPCR after 24 weeks of inoculation. *C*Las population in Shatangju was subsequently higher than in the lemon leaves, indicating that lemon exhibits more resistance against *C*Las than Shatangju (Figure 1C). To assess the difference in resistance between lemon and Shatangju, the symptomatic leaves per plant were counted in both varieties. The occurrence of symptomatic leaves in Shatangju was 23.7%, significantly greater than that in lemon (9.5%), signifying that lemon plants possess stronger resistance to HLB (Figure 1D). Throughout the scrutiny, no *C*Las was detected in mock-inoculated plant samples.

### 2.2. The Disparity in CLas Resistance between Lemon and Shatangju Is Attributed to Their Inverse Gene Expression

The Illumina HiSeq4000 platform generated a sequencing depth of 54~60 million reads per library. Q20 and Q30 percentages were above 95%, and the guanine–cytosine percentage (GC%) was calculated above 44% in each library (Appendix A). By using two-fold difference as a standard selection criterion (|log2(HLB/Mock)| ≥ 1 or ≤−1; *p*-value ≤ 0.05), 1750 Shatangju DEGs were identified, among which 779 DEGs were upregulated while 972 DE-mRNAs were found to be downregulated (Appendix A) (Figure 2A). While in lemon, a total of 3076 significant DEGs were identified, among which 2155 DEGs were upregulated and 921 DEGs were found to be downregulated (Appendix A) (Figure 2A). Lemon showed a higher percentage of upregulated DEGs (70%) compared with Shatangju (45%), indicating extensive gene reprogramming at the transcript level. A total of 15% of DEGs were shared between lemon and Shatangju, while 58% and 27% of DEGs were conserved in lemon and Shatangju, respectively (Figure 2B). Hierarchal clustering analysis was performed to determine the expression pattern of identified DEGs. Hierarchical clustering analysis revealed an altered expression pattern between *C*Las infected and mock-inoculated samples; significantly higher gene expression was observed in *C*Las infected lemon than in Shatangju (Figure 2C). The scattered and volcano plots depicted the overall expression of DEGs, with the visualization based on mean FPKM and -log 10 (*p*-value), respectively (Figure 2D,E).

### 2.3. The Inverse Gene Expression Patterns between Lemon and Shatangju Lead to Disparities in Defense-Related Pathways Regulation

The query of whether a systemic defense response against *C*Las coordinately stimulates DEGs encouraged us to conduct an enrichment analysis. GO and KEGG analyses were performed to provide a comprehensive understanding of the biological functions of DEGs. In GO analysis, DEGs were categorized based on their biological process (BP), cellular component (CC) and molecular function (MF). In BP, cellular response to stimulus (GO:0051716), carbohydrate metabolic process (GO:1901135), lipid metabolic process (GO:0006629), defense response (GO:0006952), and cell wall organization or biogenesis (GO:0071554) were found to be enriched. While in the CC, transcription factor complex (GO:0005667), cell part (GO:0044464), intracellular part (GO:0044424), cell wall (GO:00056180), and protein complex (GO:0043234) were sorted as enriched terms on the basics of their *p*-value. Moreover, in MF, binding (GO:0005488), transferase activity (GO:0016757), oxidoreductase activity (GO:0016705), and ADP binding (GO:0043531) were found to be highly enriched in the current study (Figure 3A). Identified DEGs were also linked to the KEGG database to determine the further immersion of different biological pathways. Enriched KEGG pathways were sorted, and a bubble graph was drawn (Appendix A) (Figure 3B). Overall, lemon DEGs involved in various defense-related pathways like plant–pathogen interaction, protein phosphorylation, lipid metabolism, carbohydrate metabolism, cell wall metabolism, and antioxidant activity exhibited altered and induced gene expression compared with Shatangju (Appendix A). Interestingly, lemon DEGs mainly encode RPM1, PTI1, lipoxygenase, peroxidase, LRR-RLK, PAL, RFK1, FLS2, CRKs, MYB, and WRKY transcription factors. The findings indicate that induced genes potentially play a significant role in conferring notable tolerance to HLB in lemon (Appendix A).

#### 2.3.1. Lemon Exhibits Induced Expression of DEGs Involved in Carbohydrate and Cell Wall Metabolism upon *C*Las Infection

In the realm of carbohydrate and cell wall metabolism pathways, the transcriptional activity of genes was observed to be heightened in HLB-infected lemon compared with those of the Shatangju. Briefly, lemon DEGs involved in carbohydrate metabolism mainly encode xyloglucan, G-type lectin S-receptor-like serine, pectinesterase inhibitor, BR signaling, endochitinase, and L-ascorbate oxidase. The increased expression of the mentioned genes assists lemon plants in combating HLB infection (Appendix A). In contrast, Shatangju plants exhibited induced expression of pectinesterase and callose synthase genes, whereas genes encoding endochitinase, peroxidase, and UDP-glucuronate demonstrated downregulation. These observations imply that Shatangju plants exhibit diminished antioxidant enzyme activity and heightened callose synthesis, thereby potentially facilitating susceptibility to *C*Las infection. Moreover, the burst of DEGs involved in cell wall modification, organization, and cell wall biogenesis revealed intricate patterns of expression regulation between the two varieties. In both pathways, genes encoding pectinesterase were significantly downregulated in lemons, while they were upregulated in Shatangju. These observations suggest that the heightened activity of pectinesterase and expansin-A1-like protein in the Shatangju variety makes it more prone to pathogen invasion, thereby facilitating the progression of *C*Las (Appendix A).

#### 2.3.2. Lemon Demonstrated Elevated Antioxidant Enzyme Activity Than Shatangju upon *C*Las Infection

The present study identified 16 DEGs associated with antioxidant enzyme activity in both varieties (Appendix A). In lemon plants, induced expression of peroxidase, superoxide dismutase (SOD), ascorbate peroxidase (APX), L-ascorbate oxidase, and peroxisomal catalase (CAT) aids in scavenging the ROS accumulation, thereby reducing oxidative stress caused by *C*Las infection, ultimately enhancing lemon’s tolerance to HLB. Elevated ROS levels can increase malondialdehyde (MDA) production, which serves as a marker for heightened lipid peroxidation and oxidative stress within cells. Interestingly, wet-lab experiments confirmed that Shatangju plants exhibit a higher accumulation of H_2_O_2_ (44.5%) and MDA (65.2%) compared with lemon, indicating greater oxidative stress in Shatangju (Figure 4A,B). A higher level of ion leakage was also observed in Shatangju plants than in lemon (Figure 4C). On the contrary, lemon plants exhibit higher activity of antioxidant enzymes than Shatangju. Specifically, the activity of CAT was increased by 64%, APX by 35% and SOD by 30% in lemon plants than Shatangju (Figure 4D,E). The elevated level of antioxidant activity in lemon plants, coupled with increased levels of H_2_O_2_, MDA, and ion leakage in Shatangju, contributes to the varying tolerance levels between the two varieties.

### 2.4. The Difference in Starch and Callose Accumulation between Lemon and Shatangju Is Directly Influenced by CLas Infection

Light microscopic analysis revealed anatomical disparities between the midribs of Shatangju and lemon leaves. *C*Las infected Shatangju plant samples showed thickening and deformation of the phloem, deposition of starch grains, and other disease symptoms compared with mock samples. Significant starch accumulation was evident in phloem parenchyma cells from infected Shatangju leaves, whereas such accumulation was not observed in infected lemon leaves (Figure 5A). Interestingly, mock samples had phloem cells that were smaller on the inside and larger on the outside, in contrast, HLB-affected phloem cells exhibited uniform sizes, signifying a substantial disruption in phloem development, hindering the maturation process of phloem cells. Moreover, a significant difference in starch content was observed in the HLB-infected leaves of Shatangju and lemon plants (Figure 5B). In addition to starch accumulation, callose deposition was also quantified in both plants. Noticeable callose deposition was observed in sieve elements from the HLB-infected midrib of Shatangju but not from the mock samples (Figure 6A). Quantitative analysis revealed that the callose contents in the HLB-infected midribs of Shatangju and lemon were eleven and six times greater, respectively, compared with mock samples. Specifically, HLB-infected Shatangju exhibited significantly higher callose deposition in midribs than infected lemon by more than two-fold (Figure 6B).

### 2.5. Significantly Highly Differentially Expressed Genes Mitigate HLB More Effectively in Lemon Than in Shatangju

Next, we focused on the significantly differentially highly expressed Shatangju and lemon genes to see which specific genes are altered by *C*Las infection. After cut-off (total FPKM ≥ 1000), we retrieved 28 and 26 highly expressed genes in lemon and Shatangju, respectively. In lemon, most of the upregulated genes encode disease resistance proteins, for example, c38630_g1, c49385_g1, c53637_g1, c41243_g2, c32734_g1, and c50705_g1 c28962_g1, which were annotated for leucine-rich repeat receptor-like protein, ethylene-responsive transcription factor ERF107, endochitinase CH25, sucrose synthase, L-ascorbate peroxidase, serine/threonine-protein kinase, and MAP kinase-interacting serine/threonine-protein kinase, respectively. These upregulated and highly expressed genes are indicative of their potential role in boosting lemon’s resistance against *C*Las infection. (Appendix A). On the contrary, in Shatangju, a majority of the disease resistance proteins encoding DEGs were found to be downregulated, i.e., c112328_g1, c51161_g1, c83359_g1, c69428_g1, c64163_g2, c80647_g1 and c62562_g1 which encodes a lectin-related protein, inhibitor of trypsin and Hageman factor, LRR receptor-like serine/threonine protein kinase, glycine-rich cell wall structural protein, MAP kinase-interacting serine/threonine protein kinase 1, oxidoreductase activity, and superoxide dismutase, respectively (Appendix A). The downregulation of these defense-related DEGs indicates the higher susceptibility of Shatangju against *C*Las infection.

### 2.6. An Overview of sRNA Profiles and Comparative Expression Analysis of Conserved miRNAs

An average of 16~26 million raw reads were produced in each sRNA library. Bases with poor 3′ end sequences, adapter sequences, self-ligated adapters, reads with N-containing ratios above 10%, and short reads (<18 nt) were trimmed. Quality control analysis recovered 13~16 million clean reads per library (Appendix A). Clean reads were annotated to the Rfam database and classified into different categories and their abundance was assessed, as shown in the pie chart (Appendix A). Following QC, clean reads underwent length distribution analysis. It was observed that most of the abundant reads were 21 nt long, followed by 22 and 24 nt in length (Appendix A). The boxplots displayed the normalized log10 (FPKM + 1) values for miRNA. The boxplots showed a wider spread of miRNA expression in *C*Las infected samples than in mocked samples (Appendix A). Next, pairwise compression analysis revealed the total number of conserved miRNAs in four comparable groups. In the comparison group, stj-HLB vs. stj-mock showed 21 and 33 up and downregulated miRNAs, respectively. Similarly, the Lm-HLB vs. Lm-mock comparison revealed 36 up and 18 downregulated miRNAs. In the stj-HLB vs. Lm-HLB comparison, 38 and 16 were up-and-down-regulated, respectively. Lastly, the stj-mock vs. Lm-mock, comparison revealed 27 up and 27 downregulated miRNAs (Figure 7A). The Venn diagram represents the unique and common differentially expressed conserved miRNAs among four data sets (Figure 7B). Clustering analysis was performed to overview the expression of conserved miRNAs in Shatangju and lemon. Upon *C*Las infection cluster analysis revealed the altered expression of miRNAs. Briefly, csi-miR408, csi-miR398, csi-miR393, csi-miR166, and csi-miR3954 were downregulated in lemon but found to be upregulated in Shatangju miRNAome. On the contrary, expression levels of csi-miR399, csi-miR396, and csi-miR159 increased considerably in lemon than in Shatangju (Figure 7C).

### 2.7. miRNAs Actively Coordinate the Interaction between Citrus and CLas

Choosing miRNAs that exhibit (|log2(HLB/Mock)| ≥ 1, FPKM > 100) would be exciting to determine which genes are regulated by these DE-miRNAs. After this selection criteria, a total of 10 and 15 DE-miRNAs were identified in Shatangju and lemon, respectively. These DE-miRNAs are expected to affect the HLB response since they are abundant in both varieties. Based on the base-paring status between the queries and the miRNAs, a bioinformatic algorithm was used to predict potential target genes regulated by DE-miRNAs. Totals of 81 and 211 putative target genes were predicted for Shatangju and lemon, respectively (Appendix A). Among the DE-miRNAs, we identified 5 miRNAs that were present in both Shatangju and lemon. In this study, we denoted these miRNAs as ‘conjoint-miRNAs’. Among the conjoint miRNAs, only one miRNA (csi-miR166) was upregulated in Shatangju, while others (csi-miR162-3p, csi-miR403, csi-miR393, and csi-miR399) were downregulated in Shatangju but exhibited upregulation in lemon (Table 1).

The expression analysis of DE-miRNA-regulated target genes showed that each gene expressed oppositely to its corresponding miRNA. We aimed to elucidate the roles of csi-miR166 and csi-miR399 in both citrus varieties. The former was the only DE-miRNA that showed downregulation in lemon, while the latter caught our interest due to its distinctively expressed behavior. The function of csi-miR399 in plants has likewise been the focus of much research. In lemon, expression of csi-miR399 was induced in *C*Las infected plants, and six putative target genes were predicted. The increased expression of csi-miR399 in lemon plants reduces the expression of the E2 ubiquitin-conjugating enzyme, helping them better withstand Pi shortage. The downregulation of csi-miR399 in Shatangju increased the expression of the E2 ubiquitin-conjugating enzyme, which may ultimately lead to phosphorus deficiency, exacerbating HLB symptoms (Table 2).

However, csi-miR166 has been extensively documented in numerous studies due to its varied function in plant-microbe interaction. In this study, csi-miR166 targets three genes in lemon annotated for disease resistance protein RFL1, LRR receptor-like serine/threonine-protein kinase, and transcriptional regulator ATRX. Boosted expression of these genes could enhance lemon’s defense against *C*Las infection (Table 2). In contrast, five putative target genes which encode serine/threonine-protein kinase, pentatricopeptide-repeat-containing protein, G-type lectin S-receptor-like serine, and putative disease resistance RPP13-like protein were identified in Shatangju. In Shatangju, reduced expression of these genes may lead to a weakened defense response against *C*Las infection. Hence, these findings indicate that these DE-miRNAs may significantly guide citrus and lemons’ defense response against *C*Las infection.

### 2.8. Comparative Transcriptome and miRNAome Analysis Offer Insights into the Primary Immune Responses and Regulome

Next, we compared the transcriptome and miRNAome profiles between *C*Las infected and mock-inoculated Shatangju and lemon. We selected the transcripts that exhibited significance but were opposite to the expression of respective miRNAs. After applying the criteria (log_2_FC miRNA ^[HLB/mock]^ ≥ 1 and log_2_FC mRNA ^[HLB/mock]^ ≤ −1 or log_2_FC miRNA ^[HLB/mock]^ ≤ −1 and log_2_FC mRNA ^[HLB/mock]^ ≥ 1), we retrieved 16 and 18 miRNA/mRNA pairs in lemon and Shatangju, respectively (Appendix A). A total of 12 Shatangju and lemon *C*Las induced DE-mRNAs that showed a substantial regulator–client relationship with DE-miRNA ((log_2_FC ^miRNA (HLB/mock)^ ≤ −1 AND log_2_FC ^(mRNA (HLB/mock)^ ≥ 1) were chosen for further investigation. Annotation of these induced DE-mRNAs showed that most of these genes belong to different biological processes like ‘antioxidant activity’, ‘carbohydrate metabolism’, and ‘cell wall metabolism’. These biological processes strongly correlated with our findings revealed through transcriptome analysis. For instance, DE-mRNA c40646_g1 exhibited an antagonistic relationship with cs-miR3948, and induced expression of c40646_g1 benefited the lemon plant by enhancing its glycine-rich cell wall structure, making it harder for *C*Las to invade. Taken together, comparing the transcriptome and miRNAome allows us to pinpoint defense-related responses and comprehend the regulation of biological processes. To validate the accuracy and reliability of the RNA-seq and sRNAome data, a total of 30 genes (a few randomly selected DEGs involved in enriched KEGG pathways, and a few highly expressed genes) were chosen to validate their expression (Figure 8A,B). Two DE-miRNAs (csi-miR399 and csi-miR4166), together with their target genes, were also selected for validation by stem-loop real-time RT-qPCR (Figure 8C,D). The RNA-seq and sRNAome results are consistent with the validation experiments. The outcome indicated an accurate correlation between transcriptome and sRNAome.

## 3. Discussion

### 3.1. CLas Elicits Immune Response in Both HLB-Resistant Lemon and HLB-Susceptible Shatangju

Nearly all commercial citrus types are vulnerable to HLB, with only a few exceptions, like lemon [14]. There is ample evidence that citrus can mount an immune response to HLB [20]. Besides the lemon, a few other citrus cultivars like Kaffir lime (*Citrus hystrix*) and Mexican lime (*Citrus aurantifolia*) [15], *Microcitrus australis*, *Swinglea glutinosa*, and *M. papuana* [21] exhibit a certain level of tolerance against *C*Las infection. Varied tolerance across different citrus cultivars may be attributable to either inherent or induced immune responses in each variety [22]. *C*Las infection triggers the expression of RLK, LRR, and MAPK pathways and PR genes in lemon and Shatangju. The elicitation of the mentioned genes is the distinctive feature of PTI and ETI [23]. *C*Las infected citrus does not promptly exhibit signs of infection, and it typically takes several months for symptoms to become apparent, possibly because of citrus’s innate immune response [24]. We hypothesized that if lemon and Shatangju exhibit varying tolerance levels against *C*Las infection, we must observe altered gene expression at the transcript level in both varieties. This experimental approach allowed us to identify critical immune regulators and innate defensive responses. Functional enrichment analysis of DEGs showed that the activation of genes related to carbohydrate metabolism, cell wall metabolism, antioxidant activity, plant–pathogen interaction, protein phosphorylation, and lipid metabolism, were crucial for the tolerance of *C. limon* (Appendix A). Plant cell walls contain PRRs that detect conserved microbial molecules. Upon recognizing PAMPs, plants activate defense responses, including reinforcing cell walls and producing antimicrobial compounds, to defend against potential pathogens [25]. Pectinesterase is involved in modifying pectin, a major component of the plant cell wall. By catalyzing the de-esterification of pectin, pectinesterase can alter the structural properties of the cell wall [26]. Expansin-A1-like proteins aid in cell wall loosening by breaking noncovalent connections between cellulose microfibrils and hemicelluloses. In this study, reduced expression of the expansin-A1-like in lemon contributes to the challenge of *C*Las invasion, whereas increased expression in Shatangju might make it more susceptible to CLas infection.

Moreover, tolerance to HLB in *C. limon* was due to the improved plant defense, increased antioxidant activity, and inhibition of pectinesterase. In contrast, susceptibility to HLB was attributed to the inadequate defensive response, combined with protein phosphorylation, lipid metabolism, and lower expression of genes related to antioxidant activity in Shatangju.

### 3.2. Differential Modulation of Antioxidant Activity in Response to CLas Infection Leads to Varied Levels of Tolerance among Citrus

Oxidative stress occurs due to excessive generation and accumulation of reactive oxygen species (ROS). An imbalance between the formation and detoxification of ROS may be a potential reason for excessive ROS production [27]. The antioxidant enzymes work together within plant cells to scavenge ROS, maintain redox homeostasis, and protect cellular components from oxidative damage during stress conditions [28]. *C*Las can induce excessive ROS accumulation in citrus by stimulating the expression of RbohC, RbohF, and RbohB [29]. In this study, it was found that upon *C*Las infection, most of the lemon DEGs involved in antioxidant activity (peroxidase, SOD, APX, and L-ascorbate oxidase) were upregulated while in Shatangju suppression of these genes was observed. The transcriptomic observations were substantiated through wet-lab experiments, including quantification of ROS generation, MDA content, ion leakage, and activity of antioxidant enzymes. ROS (·OH) can induce ion leakage by disrupting cell membrane integrity. This disruption can lead to programmed cell death (PCD) due to the loss of cellular homeostasis and functionality. The findings demonstrated a higher accumulation of ROS, MDA, and ion leakage in Shatangju, whereas lemon plants exhibited elevated activity of antioxidant enzymes. These observations suggest a robust immune response in lemon and a weak antioxidant-related defense response in Shatangju. MDA is regarded as the ultimate product of membrane lipid peroxidation and serves as a broad indicator of damage caused by membrane lipid peroxidation. We observed a significant increase in MDA levels in the leaves of HLB-infected plants compared with uninfected plants. Moreover, the MDA content in diseased leaves of Shatangju was notably higher than that in diseased lemon leaves, consistent with findings from a previous study [30].

### 3.3. CLas Infection Induces Changes in Starch and Callose Deposition

HLB could potentially disrupt the sieve pore structure in the phloem of citrus trees. This disruption may be triggered by the deposition of callose in the sieve pore, which inhibits the channel for starch transport. As a result, there is an abnormal deposition of starch, leading to damage in the host cell or its subcellular structure [31]. In our study, microscopy analysis revealed that *C*Las infected Shatangju and lemon plants exhibit more starch content than mock plants. Moreover, there was a significant difference in starch accumulation between Shatangju and lemon plants. *C*Las infected Shatangju plants exhibited phloem thickening, starch, and callose compared with mock samples. Starch accumulation was significant in Shatangju leaves’ phloem parenchyma cells, whereas it was absent in lemon leaves. Mock samples showed phloem cells smaller on the inside and larger on the outside, while *C*Las infected phloem cells displayed uniform sizes, indicating a significant disruption in phloem development, impeding the maturation process. These findings were consistent with previous studies [12,32].

### 3.4. Integrative Investigation of Transcriptome and miRNAome Identified Immune Regulators in Shatangju and Lemon against HLB

By conducting an integrative analysis of the transcriptome and miRNAome, we can reveal significant immunological responses and regulome. Identifying gene expression regulators is laborious due to the regulator–client relationship’s frequently indirect and subtle nature. In contrast, the relationship between small RNAs and target genes is highly predictable [33]. We utilized the apparent relationship between miRNAs and target genes to focus on miRNA–mRNA pairings that showed considerable opposite expression. At the transcript level upon *C*Las infection, we identified different biological processes such as plant–pathogen interaction, starch and sucrose metabolism, and cell wall and lipid metabolic processes that were highly induced in lemon but reduced in Shatangju. Many other authors have already reported these enriched biological processes [20,34], supporting that our findings are reliable. When we compared miRNAome with mRNA transcriptome, we found that citrus miRNA regulome predominantly influences these critical biological processes. Our investigation of the miRNAome revealed that miRNAs play a crucial role in regulating the expression of Shatangju and lemon genes upon *C*Las infection. Taken together, the evidence from the DE-miRNAs indicates that lemon exhibits a stronger tolerance to specific biotic challenges and has a more effective nutrient uptake ability than Shatangju. The tolerance of lemon to HLB may not depend on a particular pathway or single regulator but rather on a combination of regulators that enhance defense response, improve nutrient uptake, and promote the production of beneficial metabolic products that support the host or hinder pathogen survival. Further molecular genetic techniques and functional studies are needed to validate the HLB-resistant phenotype when applying the regulators.

## 4. Materials and Methods

### 4.1. Plants Material

One-year-old seedlings of Shatangju mandarin (*Citrus reticulata* Blanco cv ‘Shatangju’) and lemon (*Citrus limon* (L.) Burm. F.) in April 2023 were grafted with *C*Las infected and *C*Las-free buds. The department of Qingyuan National Citrus Seedling Breeding Base (Guangdong, China) provided vigorous Shatangju and lemon seedlings, which were grown as part of a certification program. Buds from disease-free, healthy citrus plants were used to mock-inoculate the control plants. Each plant was kept in a greenhouse in the same temperature-controlled environment (25–28 °C). Fluorescence quantitative PCR was used to detect whether the seedlings had been infected with *C*Las. Mock- and *C*Las infected Shatangju and lemon plants, exhibiting comparable disease progression, were utilized for sRNA and RNA-seq library construction. After 24 weeks of inoculation with the appearance of typical HLB disease symptoms, mature leaves from both varieties were individually collected from four different orientations of five *C*Las-positive and *C*Las-free trees. Two leaves were collected from each position. The collected samples were promptly frozen in liquid nitrogen using properly labeled tubes and then ground using mortar and pestle in liquid nitrogen.

### 4.2. RNA Extraction and RNA-Seq Libraries Preparation

Due to the asymmetrical distribution of *C*Las among citrus trees [35], it is tricky to identify whether all inoculated plants contain *C*Las. Therefore, we employed quantitative real-time PCR results to ensure the presence of *C*Las in both varieties. Mock- and *C*Las infected leaves were used for total RNA extraction as per the manufacturer’s instruction (Invitrogen, CLD, CA). The RNA content was measured after the elimination of genomic DNA using DNase I (Takara, Japan). The RNA integrity number (RIN ≥ 6.5) was evaluated using the Bioanalyzer 2100 (Agilent) and subsequently quantified using the ND-2000 (NanoDrop Technologies). Samples with high RNA quality (OD260/280 = 1.8~2.2, OD260/230 ≥ 2.0, RIN ≥ 6.5, 28S:18S ≥ 1.0, >10 μg) were used for sequencing library construction by using the TruSeq Small RNA Library Prep Kit. RNA-seq libraries were generated following the TruSeq^TM^ RNA sample preparation kit (Illumina, San Diego, CA, USA). 5 μg of RNA was used to create RNA-seq transcriptomic libraries. Messenger RNA (mRNA) was isolated using polyA selection using oligo(dT) beads and subsequently fragmented using a fragmentation buffer. The process of adding and joining the Illumina-indexed adaptors was carried out following the protocol provided by Illumina. The libraries were later chosen based on the size of cDNA target fragments, specifically, those measuring 200–300 bp, using a 2% low-range ultra-agarose. These selected libraries were then subjected to PCR amplification using Phusion DNA polymerase (NEB) for 15 PCR cycles. The TBS380 quantified the samples, and then the Illumina HiSeq 4000 sequenced the paired-end libraries.

### 4.3. Data Processing and Analyzing

Following Illumina sequencing of Shatangju and lemon samples, the resulting sequences underwent adapter clipping and quality filtering. The initial step was the purification of raw sequencing reads. Indeterminate remnants were removed from both ends of the sequence. Bases with a Phred quality score below 20 were eliminated from the sequence’s 3ʹ end. Noncoding RNA sequences, such as rRNA, tRNA, snoRNA, and snRNA, were eliminated by conducting BLASTN similarity searches against the Rfam database (Release 9.1, http://rfam.sanger.ac.uk/, assessed on 10 December 2023). The trimming and quality checking of the paired-end reads were executed using SeqPrep (https://github.com/jstjohn/SeqPrep, assessed on 10 December 2023) and Sickle (https://github.com/najoshi/sickle, assessed on 10 December 2023), respectively, with the default settings. The transcriptome was de-novo assembled using Trinity (V 2.1.1) [36]. We used Bowtie to align our RNA-seq data with the transcriptome, and the mapping percentage was approximately 81%. The mapping criteria for bowtie were defined as follows: sequencing reads must have a unique match to the genome, allowing a maximum of 2 mismatches and no insertions or deletions. Then, we estimated the levels of gene expression from our RNA-seq data using RNA-seq by Expectation-Maximization (RSEM).

### 4.4. Differential Expressional and Functional Enrichment Analysis of DEGs

EdgeR was used to perform downstream statistical analysis, such as differentially expressed genes (DEGs) identification. To determine whether transcripts showed differential expression between the samples, we used the fragments per kilobase of exon per million mapped reads (FRKM) approach to determine the expression level for each transcript. Presumptive DEGs between two samples were chosen based on the following criteria: i) a *p*-value of less than 0.05 and a logarithmic fold change of more than 1 but less than −1. Based on the same dataset, a clustering analysis was also carried out with the help of the hclust package in the R package (http://www.R-project.org, assessed on 12 December 2023). The Blast2GO software produced Gene Ontology (GO) annotations for the DEGs. The DEGs were additionally annotated in the Kyoto Encyclopedia of Genes and Genomes (KEGG) database using the KEGG automatic annotation server. The functional enrichment of the DEGs was analyzed using Fisher’s exact test, with the Benjamini and Hochberg (BH) correction applied for statistical significance [37]. Meanwhile, the KEGG pathway enrichment analysis was also assessed.

### 4.5. Small RNAs Library Construction

To investigate the expression of lemon and Shatangju sRNAs in response to *C*Las infection, total RNA was extracted from *C*Las infected and mock leaves, and eight libraries were constructed successfully. The construction of small RNA libraries was performed using the TruSeq Small RNA Library Prep Kit, following the instructions provided by the manufacturer (Illumina, San Diego CA, USA). After sequencing, the raw data contain splices and low-quality sequences. Raw data were processed and filtered by using cutadapt and trimmomatic Version 0.36 software. Initially, sequences containing the N base were eliminated. Subsequently, low-quality and splice sequences were removed from the reads. Moreover, reads shorter than 18 nt or longer than 32 nt were discarded, yielding clean data. Lastly, repetitive sequences were eliminated, and unique reads were obtained. These final high-quality data will be utilized for succeeding analyses.

### 4.6. Identification of Different Classes of Small-RNA

Once clean reads were obtained, reads with identical sequences were merged to create unique sequences. These sequences were then used to count the number and types of small RNAs in the sample and the distribution of common and unique small RNA sequences across all samples. The Rfam (http://Rfam.sanger.ac.uk/, assessed on 2 January 2024) database was used to annotate the measured small RNAs and exclude non-miRNA sequences, such as rRNA, snoRNA, and tRNA. The preprocessed sRNA sequences were annotated to the reference genome by using Bowtie, permitting a maximum of two mismatches. To find conserved microRNAs, the sRNA candidates were subjected to a BLAST search against the microRNA database available at (http://www.mirbase.org/, assessed on 2 January 2024), and miRDeep2 was used to determine conserved miRNAs.

### 4.7. Prediction of Target Genes

The plant miRNAs demonstrated near-perfect complementarity with their target mRNAs. miRNAs can attach themselves to various parts of the target mRNA. This attachment can result in either the cleavage of the target mRNA or the inhibition of its translation inside the plant. The prediction of miRNA targets was performed using the TargetFinder program (https://github.com/carringtonlab/TargetFinder/, assessed on 4 January 2024) with default parameter settings. In short, possible targets identified via FASTA searches were evaluated employing the position-dependent mispair penalties scheme with specific scoring parameters. Penalties were given for bulges, mismatches, and gaps. If a bulge, mismatch, gap, or G:U pair happened at positions 2 to 13 from the 5′ end of the microRNA, the penalties were doubled. Only one single-nucleotide bulge or single-nucleotide gap was permitted. Only anticipated microRNA targets with scores of four from a validated reference set were deemed acceptable. The functional annotation for known microRNA targets was conducted using the NCBI nonredundant protein database.

## 5. Conclusions

Transcriptomic comparison between HLB-tolerant lemon and HLB-susceptible Shatangju reveals insights into the dynamics of the host response to HLB. *C*Las infection triggers immune-mediated response in both varieties which results from excessive ROS production and starch and callose accumulation in phloem tissues. Lemons exhibit a remarkable ability for the fine-tuning of gene expression involved in cell wall metabolism and scavenging ROS, which might contribute to its tolerance to HLB. However, the Shatangju vulnerability to HLB might be attributed to its weak defense response, low antioxidant enzyme activity, and the induction of expansin-A1 proteins and pectinesterase. While comprehensive sRNAome analysis discovered the active role of DE-miRNAs in lemon and Shatangju. Altered regulation of miRNAs (csi-miR399, csi-miR166 in lemon and Shatangju may contribute to attributed HLB symptoms. This study sheds light on the pathogenicity mechanism of the HLB pathosystem and tolerance mechanism against HLB, and it provides valuable insights into HLB management (Appendix A). However, biological systems are inherently complex, and bioinformatic analyses may oversimplify or overlook certain aspects of biological processes. For instance, there was a slight decrease in the expression of genes related to antioxidant enzymes in *C*Las infected lemons, but this did not ultimately impact the reduction in final reactive oxygen species in lemons.

## Figures and Tables

**Figure 1 plants-13-01496-f001:**
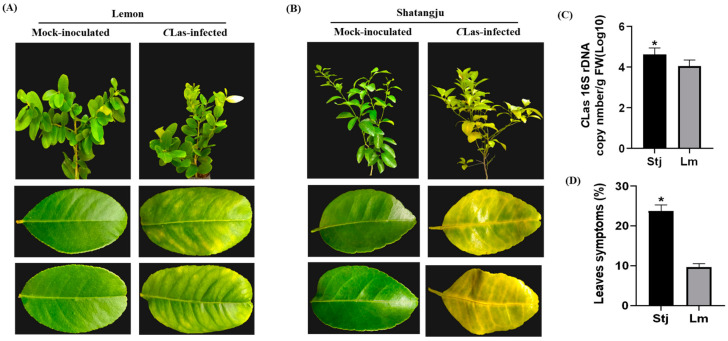
Comparative phenotypical analysis of HLB resistance between lemon and Shatangju in the greenhouse. (**A**,**B**) Phenotypical representation of mock-inoculated and *C*Las infected lemon and Shatangju trees and their leaves after 24 weeks of *C*Las infection. (**C**) Quantitative assessment of *C*Las populations in Shatangju and lemon after 24 weeks of CLas infection. (**D**) Comparative examination of the disease frequency between Shatangju and lemon following *C*Las infection. The percentage of infected leaves was calculated. Each bar represents the average of 10 plants for each variety. Standard errors were computed based on data collected from 10 plants per variety across three separate trials. Error bars represent the standard error of the means, and asterisk * represents the significant difference between lemon and Shatangju. There is a notable distinction between Shatangju and lemon according to Tukey’s test (*p* < 0.05). Stj = Shatangju, Lm = Lemon.

**Figure 2 plants-13-01496-f002:**
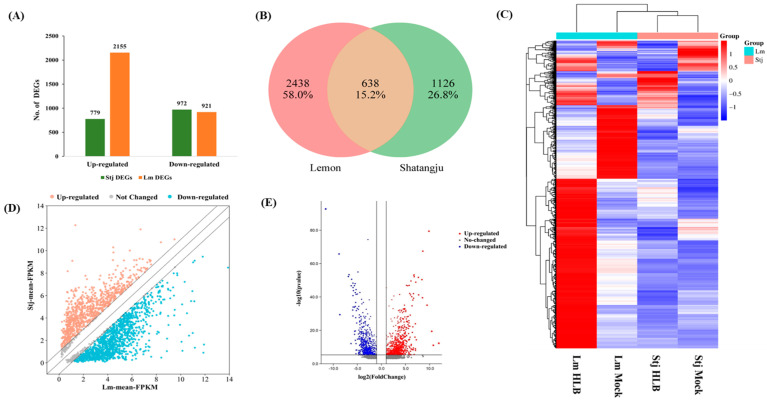
Identification of significantly differentially expressed genes (DEGs). (**A**) Number of DEGs (upregulated/downregulated) between citrus and lemon DEGs upon *C*Las infection. (**B**) Venn diagram elucidates the conserved and common DEGs between *C*Las infected and mock-inoculated Shatangju and lemon. (**C**) Cluster analysis of identified DEGs showing the expression trend between Shatangju and lemon upon *C*Las infection; the heatmap was made based on Log2FC values. (**D**,**E**) The scatter and volcano plot represent the overall distribution of DEGs between Shatangju and lemon upon infection. Scattered and volcano were plotted using mean-FPKM and log2FC values, respectively. Stj = Shatangju, Lm = Lemon.

**Figure 3 plants-13-01496-f003:**
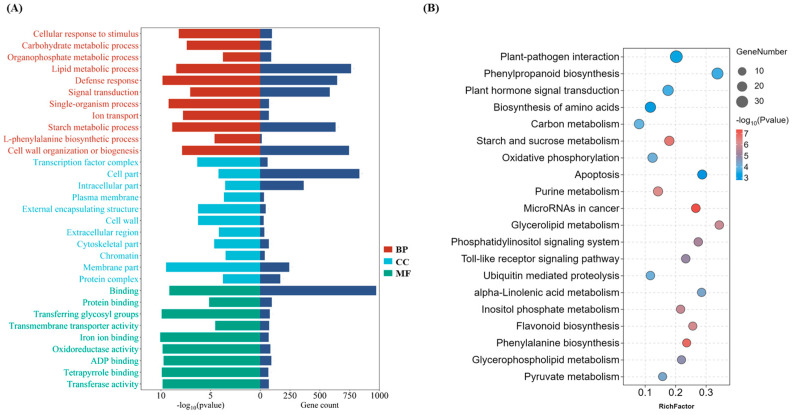
Enrichment (GO and KEGG) analysis of lemon and Shatangju DEGs. (**A**) Gene Ontology (GO) analysis was conducted to identify genes associated with specific functional categories, including biological processes (BP), cellular components (CC), and molecular functions (MF). The genes were categorized based on their roles and relationships within these distinct functional domains. (**B**) Enriched KEGG pathways analysis. The bubble plot displays the top enriched pathways, providing gene count and rich factor information for each KEGG pathway.

**Figure 4 plants-13-01496-f004:**
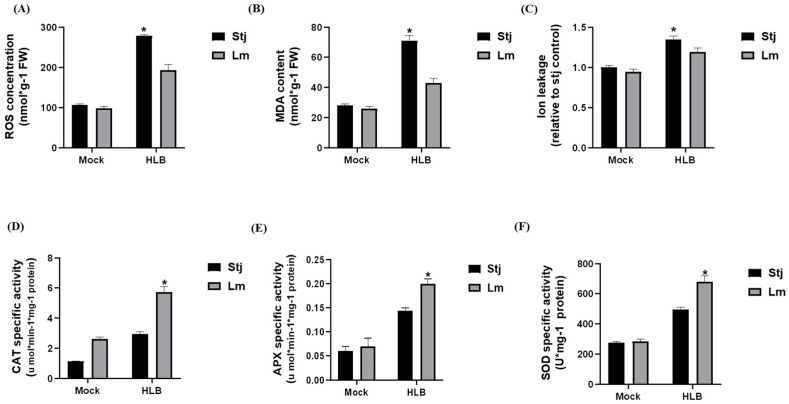
The impact of HLB infection on the biochemical components in Shatangju and lemon. (**A**) ROS concentration, (**B**) MDA content, (**C**) Ion leakage, (**D**) CAT activity, (**E**) APX activity, and (**F**) SOD activity. Asterisks * on bars represent significant differences at *p*-value (<0.05). Stj = Shatangju, Lm = lemon.

**Figure 5 plants-13-01496-f005:**
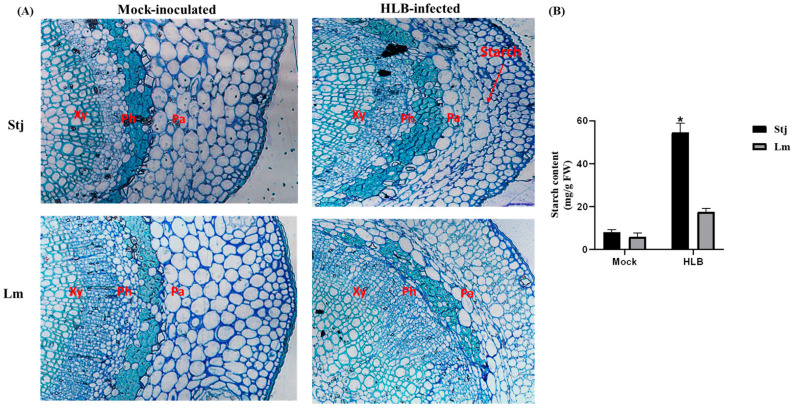
Starch accumulation in Shatangju and lemon leaves. (**A**) Microscopic analysis of phloem tissues of *C*Las infected Shatangju and lemon plants. Midribs were obtained from leaves displaying characteristic symptoms, as depicted in Figure 1A. More starch grains can be observed in *C*Las infected samples than in mock. (**B**) Starch content in leaf tissue of Shatangju and lemon collected after 24 weeks of *C*Las infection. The vertical bars reflect the standard deviations of the means of the three replicates. The asterisk (*) indicates a significant difference between Shatangju and lemon, as determined by Tukey’s test (*p* < 0.05). Stj = Shatangju, Lm = lemon.

**Figure 6 plants-13-01496-f006:**
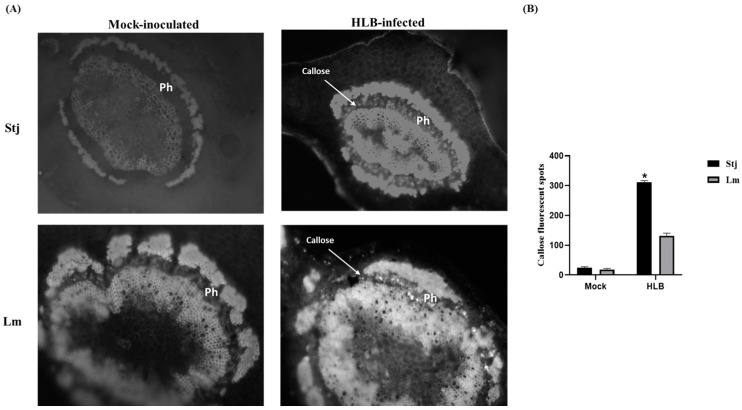
Callose deposition in leaf midribs. (**A**) Anatomical analysis of callose in midribs phloem tissues of Shatangju and lemon 24 weeks after *C*Las infection. Midribs were collected from leaves with prominent HLB disease symptoms as shown in (Figure 1A,B). Light areas signify callose deposits in the phloem tissues. (**B**) Measurement of callose contents in the midribs of leaves 24 weeks after *C*Las infection. The quantification of callose was determined by counting the number of fluorescent callose spots present in the phloem of each sample. The vertical bars reflect the standard deviations of the means of the three tests. The asterisk (*) indicates a significant difference between Shatangju and lemon, as determined by Tukey’s test (*p* < 0.05). Stj = Shatangju, Lm = lemon.

**Figure 7 plants-13-01496-f007:**
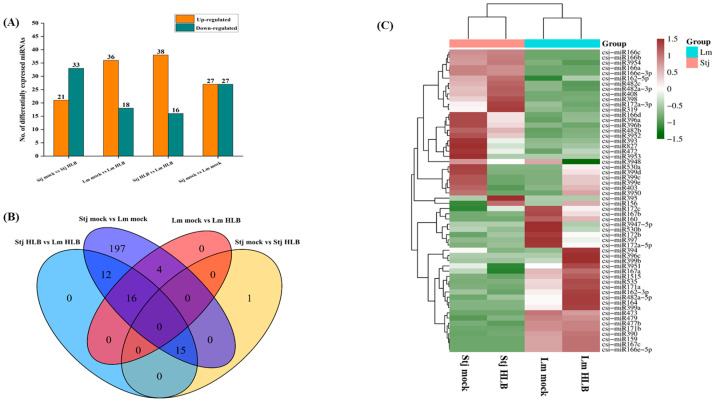
Comparative analysis of known miRNAs and their expression profiles in Shatangju and lemon. (**A**) The bar chart shows the number of upregulated and downregulated conserved miRNAs in four compared data sets. The X-axis represents the compared group and the Y-axis indicates the number of differentially expressed miRNA (DE-miRNA). (**B**) Venn diagram elucidates the overlapped and distinct no. of conserved miRNAs in different groups, brown color represents up-regulated miRNA while green shows down-regulated. (**C**) Cluster analysis (Heatmap) of DE-miRNAs between Shatangju and lemon after *C*Las infection. Lm = lemon; Stj = Shatangju.

**Figure 8 plants-13-01496-f008:**
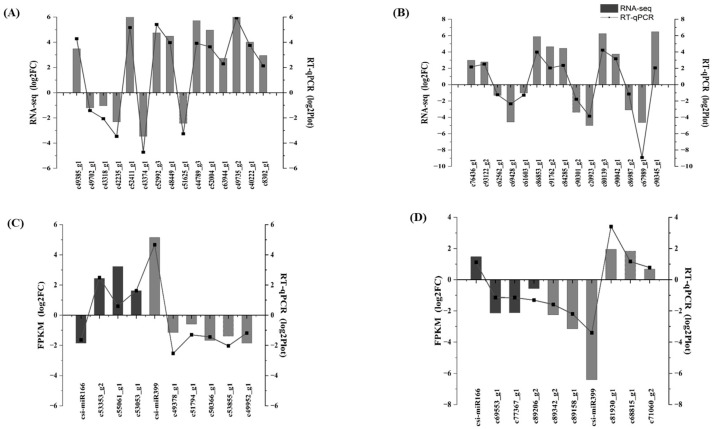
RT-qPCR validation of transcriptome and sRNAome**:** (**A**) Validation of 15 lemon DEGs associated with enriched GO terms and KEGG pathways. (**B**) Validation of 15 Shatangju DEGs associated with enriched GO terms and KEGG pathways. (**C**) RT-qPCR validation of csi-miR166 and csi-miR399 and its target genes in lemon. (**D**) RT-qPCR validation of csi-miR166 and csi-miR399 and its target genes in Shatangju. All the experiments were repeated three times.

**Table 1 plants-13-01496-t001:** List of conjoint DE-miRNAs (FPKM > 100).

No.	miRNA-ID	Length (nt)	log2FC Lm	log2FC Stj
1	csi-miR166	21	−1.85	1.47
2	csi-miR162-3p	21	2.55	−1.14
3	csi-miR403	21	3.38	−1.59
4	csi-miR393	22	2.08	−1.95
5	csi-miR399	21	5.18	−6.41

**Table 2 plants-13-01496-t002:** Expression of two conjoint DE-miRNAs along their target genes annotation.

		DE-miRNA				Transcript				
	miRNA-ID	Total FPKM	Mock	HLB	log2FC	Target Genes	Mock	HLB	log2FC	Description
Lemon	csi-miR166	2682.62	2098.73	583.89	−1.85	c53353_g2	1.03	5.56	2.43	Disease resistance protein RFL1
						c55061_g1	50.75	475.69	3.23	LRR receptor-like serine–threonine protein kinase
						c53053_g1	14.9	45.72	1.62	Transcriptional regulator ATRX
	csi-miR399	123.06	3.27	119.79	5.15	c49378_g1	36.49	16.6	−1.14	E2 ubiquitin-conjugating enzyme (PHO2)
						c51794_g1	29.92	19.94	−0.59	E2 ubiquitin-conjugating enzyme (PHO2)
						c50248_g2	145.25	57.52	−1.34	hypothetical protein CISIN_1g002895mg
						c50366_g1	65.34	20.57	−1.67	E2 ubiquitin-conjugating enzyme (PHO2)
						c53855_g1	183.25	70.08	−1.39	Plant peroxidase signature
						c49952_g1	10.26	2.84	−1.85	E2 ubiquitin-conjugating enzyme (PHO2)
Shatangju	csi-miR166	2759.05	730.44	2028.61	1.47	c77367_g1	10.28	2.37	−2.12	Pentatricopeptide-repeat-containing protein
						c89206_g2	35.35	23.98	−0.56	Signal recognition particle 54 kDa protein;
						c89342_g2	25.74	5.43	−2.24	G-type lectin S-receptor-like serine
						c89158_g1	31.25	3.54	−3.14	Putative disease resistance RPP13-like protein 3
	csi-miR399	253.76	250.92	2.84	−6.41	c81930_g1	2.71	10.68	1.94	E2 ubiquitin-conjugating enzyme (PHO2)
						c85783_g2	92.5	165.38	0.84	Alkaline/neutral invertase CINV1
						c92767_g2	57.28	98.36	0.78	Carboxypeptidase D; proteolysis;
						c68815_g1	21.24	75.36	1.83	E2 ubiquitin-conjugating enzyme (PHO2)
						c71060_g2	2.57	4.13	0.68	E2 ubiquitin-conjugating enzyme (PHO2)

## Data Availability

The raw data presented in this study are openly available in the NCBI database. small RNA data, Bio project number: PRJNA1088633. RNA-seq data, Bio project number: PRJNA1088422.

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
