# Peer review of "Comparative Transcriptome and sRNAome Analysis Suggest Coordinated Citrus Immune Responses against Huanglongbing Disease"

_plants, 2024, doi:10.3390/plants13111496_

Round 1

Reviewer 1 Report

Comments and Suggestions for Authors

This manuscript (MS) Integrative Transcriptome and sRNAome Profiling Revealed HLB-Tolerant Mechanism Between Lemon and Shatangju in Response to CLas-Infection. The authors systematically analyzed the transcriptome and sRNAome in resistant and susceptible cultivars. Overall, MS is well-written and easy to read. The presentation of the data results is adequate; however, an experiment is required to prove the claim of miRNAs and the corresponding gene targets. Further, I felt the results section needed to be concise for better quality. I recommend that acceptable with minor editing in language and provided to conducting experimental evidence to support the results section.

Comments:

1.       All the figures are to be recreated for better resolutions for better readability.  

2.       In lemon csi-miR162-3p, csi-miR403, csi-miR393, and csi-miR399 were significantly enriched compared to susceptible cultivar. The author must prove this by selecting a few of these miRNAs to overexpress and down-regulate transiently in citrus, which will support the hypothesis of results. 

Comments on the Quality of English Language

Minor editing of scientific language is required. 

Author Response

Hi,

I trust this message finds you in good health. We have addressed all of your concerns. For further details, please see the attachment.

Reviewer 2 Report

Comments and Suggestions for Authors

The authors report in this manuscript the difference in response to the HLB disease between two citrus varieties. In particular, their study was conducted on the genetic base of these varieties. Their results are expected to contribute to phytopathological research in citrus. The manuscript can be published in the journal after revision. I hope the comments below would be available for revising the manuscript.

Major issues

1. English

English should be edited by a native English speaker before the resubmission of the manuscript.

2. Tolerance/resistance

The authors need to distinguish these two biological terms. The authors report here the genes which are attributed to the mitigation of the severance of the disease in the citrus. In this context, the mechanism of the reduced symptoms in the lough lemon can be considered owing to the resistance, not tolerance. The authors also use “immune” in the text. However, the mitigation is not acquired by the variety but is just caused by the genes that are involved in suppressing the pathogen multiplication. The authors should carefully check the use of this word.

Minor issues

Clas in the title

The authors should not use abbreviations when any words are referred to first in the manuscript. The title should be changed.

L60

Both the genera Fortunella and Poncirus have been changed for Citrus, thought these genus names are still used in some papers.

L82

Citrus trifoliata is recommended to use.

L109-112

These sentences should be moved to Materials and Methods.

Figure 1

Photos and graph panels may be separated in different figures for easy understanding.

L143

Spell out “GC”.

L205-207, 215-216, 221-224 and 227-231

These descriptions should be moved to Discussion.

L234-235, L236-242 and L245-246

These descriptions should be moved to Discussion.

L268

Refer to the results of statistical analyses with not only the probability but their statistics for the significance.

L289

Refer to the results of statistical analyses for the significance.

Figures 1-3 and 6-

Some letters are too small to read. Use larger ones.

L345

Two words, likely and significantly, cannot be compatible. The word “significantly” is used when the effects of miRNAs were confirmed. This means these effects were not likely.

L349

Delete the period before “(“.

L364-366

Move the sentence to Discussion.

L374-376

Move the sentence to Discussion.

L508

Refer to the date when the grafting was performed.

L512

Describe how the samples were collected and when it was.

Comments on the Quality of English Language

English should be edited by a native English speaker before the resubmission of a revised manuscript.

Author Response

(The authors gave the same response as above.)

Round 2

Reviewer 1 Report

Comments and Suggestions for Authors

The authors addressed my comments, and the manuscript ca be acceptable for publication. 

Author Response

Dear Reviewer,

                         We wish to express our heartfelt appreciation to the esteemed reviewer who accepted our manuscript. Your recognition of the merit and significance of our work is deeply gratifying and underscores the dedication we have invested in this research endeavor. We are sincerely grateful for your valuable contribution to the peer-review process and for your pivotal role in advancing scholarly discourse in our field.

Reviewer 2 Report

Comments and Suggestions for Authors

Figure 10

Letters are too small to read. The two panels should be shown vertically. Or this figure may be shown as a supplementary figure.

Comments on the Quality of English Language

English has been well improved. However, the authors should still pay attention in spelling. For example, when they refer to cultivar, the name should be begun with the capital letter of the first letter in the name (shatangju in Key words or at L304). 

Author Response

Hi Dear Reviewer II,

                  Once again, thank you for your kind attention to our manuscript. we have accepted your suggestions, and we have revised our manuscript accordingly. Please see the attachment.
